# Combined Effects of CO_2_ Adsorption-Induced Swelling and Dehydration-Induced Shrinkage on Caprock Sealing Efficiency

**DOI:** 10.3390/ijerph192114574

**Published:** 2022-11-06

**Authors:** Xiaoji Shang, Jianguo Wang, Huimin Wang, Xiaolin Wang

**Affiliations:** 1State Key Laboratory for Geomechanics & Deep Underground Engineering, School of Mechanics and Civil Engineering, China University of Mining and Technology, Xuzhou 221116, China; 2College of Water Conservancy and Hydropower Engineering, Hohai University, Nanjing 210098, China; 3School of Engineering, University of Tasmania, Hobart 7001, Australia

**Keywords:** CO_2_ geological sequestration, brine water-CO_2_ two-phase flow, matrix dehydration, caprock swelling, permeability evolution

## Abstract

Carbon dioxide (CO_2_) may infiltrate into the caprock and displace brine water in the caprock layer. This causes two effects: one is the caprock swelling induced by the CO_2_ adsorption and the other is the caprock dehydration and shrinkage due to CO_2_–brine water two-phase flow. The competition of these two effects challenges the caprock sealing efficiency. To study the evolution mechanism of the caprock properties, a numerical model is first proposed to investigate the combined effects of CO_2_ adsorption-induced expansion and dehydration-induced shrinkage on the caprock sealing efficiency. In this model, the caprock matrix is fully saturated by brine water in its initial state and the fracture network has only a brine water–CO_2_ two-phase flow. With the diffusion of CO_2_ from the fractures into the caprock matrix, the CO_2_ sorption and matrix dehydration can alter the permeability of the caprock and affect the entry capillary pressure. Second, this numerical model is validated with a breakthrough test. The effects of the two-phase flow on the water saturation, CO_2_ adsorption on the swelling strain, and dehydration on the shrinkage strain are studied, respectively. Third, the permeability evolution mechanism in the CO_2_–brine water mixed zone is investigated. The effect of dehydration on the penetration depth is also analyzed. It is found that both the shale matrix dehydration and CO_2_ sorption-induced swelling can significantly alter the sealing efficiency of the fractured caprock.

## 1. Introduction

The potential leakage of the stored carbon dioxide from the caprock layers is an important environmental safety issue to the CO_2_ sequestration in geological formations [1,2]. The numerical simulations for the commercial-scale CO_2_ sequestration projects have evaluated the migration and interaction with the storage reservoir [3,4]. These evaluations focus on the trapping mechanisms and reservoir storage capacity [5]. The trapping mechanisms of the mineral, solubility, hydrodynamics, and structure have been studied, respectively [6,7,8,9]. Further, the effective storage capacity is estimated based on the plume formation, accumulative pressure, and other factors [10,11]. The structural trapping is usually formed by many caprock layers, which are physical barriers with a low permeability to prevent the CO_2_ from a further upwards migration. The CO_2_ accumulates gradually from the reservoir to the bottom of the caprock layer during the sequestration process [12,13,14] and interacts with the caprock layer. When the accumulation pressure exceeds the initial entry capillary pressure, CO_2_ gradually penetrates into the caprock layer [15,16] and the caprock sealing efficiency is impaired [17]. The leakage of CO_2_ after the breakthrough of the caprock layer damages the safety of the storage reservoir. Therefore, the key issue to the CO_2_ geological sequestration is the caprock sealing stability and efficiency. How the dehydration of the shale matrix alters the sealing efficiency during the CO_2_–water displacement has not been well investigated so far [8,18].

A caprock layer may be highly fractured and heterogeneous, even at the depth of 4 km [1,19]. Macropores in the fractures and micropores in the matrix make up the two-scale pore systems in a fractured caprock [16,20,21]. In these systems, the micropores in the matrix are the storage space of carbon dioxide because their extremely large internal surface area adsorbs CO_2_. These micropores are small in size and fully or partially saturated in the caprock layer [8]. The main component of the caprock is shale, whose shale matrix is of a low porosity and an ultra-low permeability [22]. Closely spaced fractures are usually assumed in the modeling of the macropore system [23,24]. These fractures are the main channels for the CO_2_–brine two-phase flow. Therefore, the combination of two-phase flow in the fractures, CO_2_ sorption and diffusion in the matrix, and the matrix dehydration makes up the main transport mechanisms [25]. Both the CO_2_ sorption and dehydration induce the swelling or shrinkage of the shale matrix and may modify the stability of the caprock sealing. 

If the caprock is water saturated, the dehydration of the matrix may impact the caprock sealing efficiency. Some water-saturated minerals are fully swelled under in situ conditions [8,26]. The water saturation is changing while the CO_2_ displaces the water in the matrix of the caprock. This change results in the shrinkage of the shale matrix, thus enlarges the aperture of the shale fractures and may enhance the caprock permeability [18,27]. On the other hand, some minerals such as clay may adsorb the CO_2_ molecules onto their internal surfaces, thus inducing the swelling of the matrix and the closing of some fractures. The swelling of the caprock matrix is contributed by these two competitions of a dehydration-induced shrinkage and a sorption-induced swelling. The competition effects promote the self-limiting or self-enhancing change in the physical and mechanical properties of caprocks throughout the multi-physical interaction process [28]. Hence, the CO_2_ sorption and dehydration in the caprock matrix are the key issues in the evaluation of the caprock sealing efficiency. 

The CO_2_–brine water two-phase flow in the caprock layer is a multi-physical process [29]. The mechanical deformation, the CO_2_–brine two-phase flow in the fractures, the gas diffusion in the caprock matrix, and the dehydration- and sorption-induced swelling are all included in this multi-physical process. The CO_2_ sorption/desorption, pore pressure, mechanical deformation or stress compaction, and geochemical reaction may be altered in these mechanical processes. The fractures are filled with the CO_2_–brine water two-phase flow. The two-phase flow may stimulate the local deformation of the caprock and change the wettability of the caprock and the entry capillary pressure of the fractures [30]. In addition, the flow channels can be modified through the CO_2_–rock interaction, thus altering the porosity and permeability of the caprock [18,31]. The CO_2_ sorption has an important impact on the stability of the caprock sealing. The results are variable with the organic compositions, types, and shale contents [32]. Furthermore, clay minerals in shale layers can adsorb some free gas. The capacity of gas adsorption varies with the clay type, volume, and type of gas itself. The adsorbed volume of gas is closely related to the pores surface area. It is found that there are similar sorption behaviors between the coal and shale. The Langmuir formula is applicable to the shales accordingly [33]. On the other hand, some minerals swell after absorbing water (hydration) and shrink after losing water (dehydration) [19,25]. Therefore, these interactions can seriously affect the height of the CO_2_ penetration upward or the efficiency of the caprock sealing, but no publication exploring this is available presently.

The combined effects of CO_2_ adsorption-induced swelling and dehydration-induced shrinkage on the caprock sealing efficiency are investigated in this paper. First, a two-phase flow in the fractures, the diffusion processes for CO_2_–brine water in the shale matrix, and the mechanical compaction process of fractured caprock are represented in a conceptual model. Then, the porosity and permeability constitutive models with the effect of the dehydration process are developed, respectively. The entry capillary pressure is also expressed mathematically. These models together constitute a new coupled multi-physical model. It is an extension of our previous model for the efficiency of the caprock sealing [16,24]. Third, the effects of the adsorption-induced swelling and dehydration-induced shrinkage as well as the stress compaction are expressed in terms of the fracture aperture change. Through the finite element method, this fully coupled model is solved numerically and verified by a breakthrough test of a fractured sample. Finally, the effects of the adsorption-induced swelling and dehydration-induced shrinkage on the self-limiting or self-enhancement within a caprock layer are numerically investigated. 

## 2. Multi-Physical Interaction Model for a Fractured Caprock Layer

### 2.1. Multi-Physical Interactions in a Fractured Caprock

The disturbance mechanisms of caprock sealing induced by a two-phase flow stimulation are complex. The pressure of CO_2_ at the bottom of the caprock layer continuously increases with the CO_2_ accumulation from the storage reservoir. If the CO_2_ pressure is lower than the sum of the reservoir pressure and entry capillary pressure [34,35], the Darcy flow does not occur. This kind of sealing is called the capillary sealing. In this sealing mode, the diffusion and flow of the dissolved gas in pore water are the main means of CO_2_ migration. The Darcy flow starts once the gas pressure is over the sum of the reservoir pressure and the entry capillary pressure [35]. In two-phase Darcy flow process, the CO_2_ displaces the pore water in the caprock layer. 

Caprocks are a mostly fractured porous media which consist of fractures and a matrix. The fracture usually has a much lower entry capillary pressure than the shale matrix. The shale caprock of water-saturated is different from the shale gas reservoir. The pore water still remains in micropores after the CO_2_ penetrates into the fractures [36]. As CO_2_ in the fracture network gradually diffuses into the shale matrix, water in the shale matrix will be replaced by CO_2_ and then enters the fractures. These two processes are usually diffusive and called sorption and dehydration, respectively. Figure 1 presents a detailed conceptual model of these mechanisms. In this conceptual model, the CO_2_–brine water two-phase flow is observed only in the fracture network. However, due to the interaction between the shale caprock and CO_2_, the CO_2_ adsorbs into the matrix and water is then produced due to the dehydration [37]. The matrix is subjected to the two actions of both the CO_2_ adsorption-induced swelling and dehydration-induced shrinkage (see Figure 2). Therefore, the multi-physical processes can be represented by this conceptual model in various time scales: (1) a CO_2_–brine water two-phase flow in the initial water-saturated fractures; (2) the propagation of gas front-induced mechanical process; (3) a CO_2_ diffusion and adsorption into the caprock matrix; (4) the water dehydration of the matrix; and (5) the geochemical reactions among CO_2_, water, and caprock [38]. The first three events are short term: the mechanical deformation occurs immediately when the effective stress has any change. The CO_2_–brine water two-phase flow in the fracture network is instantly started when the entry capillary pressure is run over. The diffusion, dehydration, and geochemical reaction processes may be long term, such as the span of geological time [39].

### 2.2. Dehydration and Shrinkage of Shale Matrix

The CO_2_–brine water displacement mechanism in a shale matrix block is depicted in Figure 3. Shale is a sedimentary rock formed by dewatering and the cementation of clay minerals. The clay minerals are water-sensitive [40]. The shale matrix swells when water flows into the matrix (called the hydration process). Inversely, the shale matrix shrinks when the water flows out of the matrix (called the dehydration process). Figure 3a presents a shale matrix block which is surrounded by fractures. Both the fractures and the matrix were assumed to have the same initial phase pressures of a CO_2_–brine water two-phase flow. The phase pressures changed continuously with the two-phase flow process in the fracture network (see Figure 3b) and caused CO_2_ to flow into the block and the water to flow out of the block.

#### 2.2.1. Water Content, Saturation, and Porosity of Shale Matrix

The porosity of shale matrix is calculated by:(1)$$\phi_{m}=\frac{V_{V}}{V_{T}}$$

The brine water saturation is defined as:(2)$$s_{wm}=\frac{V_{bw}}{V_{V}}$$

The volumetric water content W is related to the water saturation as: (3)$$W=\phi_{m}s_{bwm}$$

The water content (by weight) is the ratio of water mass to solid mass as:(4)$$W_{c}=\frac{M_{w}}{M_{s}}=S_{bwm}\frac{\phi_{m}}{1-\phi_{m}}\frac{\rho_{bw}}{\rho_{s}}$$

Equation (4) is a relationship among the porosity, water content, and water saturation. Obviously, any change in one factor may induce a change in the other two. Further, the water loss in the shale matrix (or the dehydration process) may be not only from free water but also from some bonding water [18]. At this time, the bonding water has the following mass [41]:(5)$$M_{b}=\left(\frac{A}{1+W_{c}}\right)\left(\frac{\alpha_{1}}{1-\alpha_{1}}\right)$$
where $\phi_{m}$ is the porosity of the shale, $S_{bwm}$ is the water saturation in the pores, $V_{V}$ is the pore volume, $V_{W}$ is the water volume, and $V_{T}$ is the total volume. $M_{bw}$ is the water mass and $M_{S}$ is the shale mass. $\rho_{bw}$ is the density of the water and $\rho_{s}$ is the density of the shale matrix. A and $\alpha_{1}$ are the correction factors for the total water and bonding water content contributing to a shrinkage deformation, respectively. 

This bonding water is usually regarded as a part of the solid particles. If this bonding water does not contribute to a shrinkage deformation, the water content of the shale matrix is revised as:(6)$${\bar{W}}_{c}=\frac{M_{bw}-M_{b}}{M_{s}+M_{b}}=\frac{\rho_{s}}{\rho_{bw}\left(1+W_{c}\right)}\left(W_{c}-\frac{A\alpha }{1-\alpha }\right)$$
where α is the correction factor for the bonding water content not contributing to the shrinkage deformation. 

#### 2.2.2. Dehydration-Induced Volumetric Strain of Shale Matrix

This matrix shrinkage is measured by the volumetric strain $\varepsilon_{vbw}$ through a moisture-adsorption test. A quadric function is here used for the moisture-swelling relationship as:(7)$$\varepsilon_{vbw}\left(W_{c}\right)=K_{1}W_{c}+K_{2}W_{c}^{2}$$
where $K_{1}$ and $K_{2}$ are the expansion coefficients. A typical hydration-induced swelling of the Mancos shale [42] is presented in Figure 4, where $K_{1}=0.212$ and $K_{2}=33.24$. Obviously, the swelling strain of this shale is big and Equation **(7)** is able to describe this relationship.

### 2.3. Dehydration-Induced Modifications for Porosity and Permeability Models

#### 2.3.1. Porosity Evolution in Homogeneous Shale Matrix

The porosity ratio is obtained as:(8)$$\frac{\phi }{\phi_{0}}=1+\left(1-R\right)\frac{S_{0}-S}{1+S}$$
where $R=\alpha /\phi_{0}$. $S_{0}$ and S are the effective volumetric strains in the initial and current state, respectively. They are defined as follows
(9)$$S_{0}=\varepsilon_{v0}+\frac{p_{0}}{K_{s}}-\varepsilon_{s0}-\varepsilon_{D0},\;\;\;S=\varepsilon_{v}+\frac{p}{K_{s}}-\varepsilon_{s}-\varepsilon_{D}$$
where $\varepsilon_{v}$ and $\varepsilon_{v0}$ are the current and initial volumetric strain, respectively, and $\varepsilon_{s0}$ and $\varepsilon_{s}$ are the initial and current volumetric strain induced by the sorption. $\varepsilon_{D0}$ and $\varepsilon_{D}$ are the initial and current hydration-induced volumetric strain, respectively. $\phi_{0}$ is the initial porosity and $p_{0}$ is the initial pore pressure.

#### 2.3.2. Local Fracture Strain

Both the fractures and matrix contribute to the total deformation of the fractured shale (see Figure 5).

The change in the fracture aperture is:(10)$$\Delta b=\left(s+b_{i}\right)\Delta \varepsilon_{e}-s\Delta \varepsilon_{em}$$

Equation (10) can be reformulated as: (11)$$\Delta b=b_{i}\left[1+\frac{s}{b_{i}}\left(1-\frac{\Delta \varepsilon_{em}}{\Delta \varepsilon_{e}}\right)\right]\Delta \varepsilon_{e}=b_{i}\left[1+n\frac{1-R_{m}}{\phi_{f0}}\right]\Delta \varepsilon_{e}$$

The local fracture strain is derived as:(12)$$\frac{\Delta b}{b_{i}}=\left[1+n\frac{1-R_{m}}{\phi_{f0}}\right]\Delta \varepsilon_{e}$$
where s is the fracture spacing and $\phi_{f0}$ is the initial fracture porosity. $b_{i}/s=\phi_{f0}/n$ are in an n-dimension case. $\varepsilon_{e}$ is the average effective strain and $\Delta \varepsilon_{e}$ is its increment in a fixed representative length ($\Delta \varepsilon_{e}=\left(S_{0}-S\right)/\left(1+S\right)$). $\varepsilon_{em}$ is the effective strain, and $\Delta \varepsilon_{em}$ is the increment of $\varepsilon_{em}$. $R_{m}=\Delta \varepsilon_{em}/\Delta \varepsilon_{e}$ is the increment strain ratio of the average effective strain to effective strain. 

#### 2.3.3. Evolution of Permeability

The initial permeability of the fracture is expressed as the following cubic law:(13)$$k_{0}=\frac{b_{i}^{3}}{12s}$$

If fracture aperture $b_{i}$ changes due to the compaction, swelling, dehydration or their combinations, the permeability k will change as follows:(14)$$k=\frac{{\left(b_{i}+\Delta b\right)}^{3}}{12s}$$
where Δb is the change in the fracture aperture.

Combining Equation (14) with Equation (13) yields the permeability ratio of the fracture as:(15)$$\frac{k}{k_{0}}={\left(1+\left(1+n\frac{1-R_{m}}{\phi_{f0}}\right)\Delta \varepsilon_{e}\right)}^{3}$$

In two-dimensional space, this permeability ratio is:(16)$$\frac{k}{k_{0}}={\left[1+\frac{2\left(1-R_{m}\right)}{\phi_{f0}}\Delta \varepsilon_{e}\right]}^{3}$$

### 2.4. Change in Entry Capillary Pressure with Fracture Deformation

The caprock is initially water saturated. The entry capillary pressure (Appendix A) limits the CO_2_ containment capacity of the caprock and depends on both the pore geometry in the matrix and the CO_2_/brine water/rock wettability [43]. The entry capillary pressure is calculated by
(17)$$p_{e}=\frac{2\sigma \cos\left(\theta \right)}{r}$$
where r is the pore radius. σ is the interfacial tension of water, CO_2_, and caprock. θ is the contact angle to express the hydrophilic interactions.

The current aperture of a fracture is:(18)$$b=b_{i}+\Delta b$$

It is equivalent between the aperture of a fracture and the representative radius of a pore in this paper. In the calculation of the entry capillary pressure in a fracture, r=b. The entry capillary pressure of the fracture at the initial state is:(19)$$p_{ei}=\frac{2\sigma_{i}\cos\theta_{i}}{b_{i}}$$

Here, the subscript ‘*i*’ denotes the initial state. The interfacial tension and wettability are influenced by the pressure, temperature, or chemical reactions during CO_2_–brine water–rock contacting process [36]. If these physical variables do not change with the deformation, the CO_2_ entry pressure is concluded as:(20)$$p_{e}=\frac{p_{ei}}{1+\left(1+n\frac{1-R_{m}}{\phi_{f0}}\right)\Delta \varepsilon_{e}}$$

This is our new entry pressure of CO_2_ after considering the effective strain.

## 3. Mass Transfer of Two-Phase Flow between Fractures and Shale Matrix

### 3.1. Mass Transfer of CO_2_ between Fractures and Shale Matrix

The sorption rate of CO_2_ is denoted by:(21)$$Q_{m}=-\rho_{c}\rho_{ga}\frac{dm_{b}}{dt}$$
where $Q_{m}$ is the source term of mass and the minus sign ‘-’ represents the mass transfer from the shale matrix to the fractures. $m_{b}$ is the residual content of CO_2_ in the matrix at pressure p. $dm_{b}/dt$ is the exchange rate of CO_2_ between the fractures and the shale matrix and is expressed as:(22)$$\frac{dm_{b}}{dt}=-\frac{1}{\tau }\left[m_{b}-m_{e}\left(p\right)\right]$$
where τ is the diffusion time, $m_{e}$ is the gas content at the equilibrium state, and p is the pressure of the fractures.

The diffusion time is defined as:(23)$$\tau =\frac{1}{aD}$$
where a is a shape factor and D is the diffusion coefficient of CO_2_ in the matrix.

### 3.2. Dehydration due to Water Transfer between Fractures and Shale Matrix

The dehydration of the shale matrix is still assumed to follow a diffusion process [42]. In this paper, this process is described by a simplified diffusion equation as:(24)$$\frac{dm_{bw}}{dt}=-\frac{1}{\tau_{bw}}\left[m_{bw}-m_{bwe}\left(s_{bw}\right)\right]$$
where $m_{bw}$ is the mass of the water phase in the shale matrix, and $m_{bwe}$ is the water content at the equilibrium state with the water saturation $s_{bw}$ in the fractures.

The diffusion time of water in the shale matrix is denoted as:(25)$$\tau_{bw}=\frac{1}{aC}$$
where C is the water diffusion coefficient in the matrix [42]. 

The CO_2_–brine water two-phase flow displacement is complicated in a porous media [44], but can be described by following the diffusion process over spherical pellets [45]:(26)$$\frac{dm_{bw}}{dt}=\frac{15D_{bw}}{R_{bw}^{2}}\left[m_{bwe}\left(S_{bw}\right)-m_{bw}\right]$$
where $D_{bw}$ is the diffusion coefficient and $R_{bw}$ is the characteristic radius of spherical pellets. It is obvious that both Equation (**26**) and (**24**) have the same form. 

## 4. Mathematical Descriptions of Multi-Physical Processes

### 4.1. Mass Conservation Laws for CO_2_–Brine Water Two-Phase Flow in Fractures

The CO_2_–brine water two-phase flow occurs in the fractures. CO_2_ is considered as the non-wetting phase, and the brine water is considered as the wetting phase. According to the mass conservation law, the governing equations of CO_2_ and brine water are obtained as follows.

For the brine water phase:(27)$$\frac{\partial \left(\phi \rho_{bw}s_{bw}+\rho_{bw}m_{bw}\right)}{\partial t}+\nabla \cdot \left(-\frac{kk_{rbw}}{\mu_{bw}}\rho_{bw}\left(\nabla p_{bw}+\rho_{bw}g\nabla H\right)\right)=f_{bw}^{\prime }$$

Being different from the brine water, the CO_2_ has two unique features: (1) a strong compressibility in both free and supercritical states and (2) the existence of both free and absorbed phases in the fractures. Furthermore, the CO_2_ diffuses from the fractures into the matrix and displaces the pore water in the shale matrix. The mass of CO_2_, $m_{nw}$, can be derived as
(28)$$m_{nw}=\phi \rho_{nw}S_{nw}+\rho_{nwa}\rho_{c}\frac{V_{L}p*}{p_{L}+p*}+\rho_{nwa}\rho_{c}m_{b}$$

Equation (28) shows that the storage states of the CO_2_ phase include a free-phase form (the first term), an absorbed form in the fracture network (the second term), and the mass exchange from the matrix (the third term).

For the CO_2_ phase:(29)$$\frac{\partial m_{nw}}{\partial t}+\nabla \cdot \left(-\frac{kk_{rnw}}{\mu_{nw}}\rho_{nw}\left(\nabla p_{nw}+\rho_{nw}g\nabla H\right)\right)=f_{nw}^{\prime }$$
where $\rho_{bw}$ is the density of the brine water. k is the absolute permeability of the shale. $k_{rbw}$ and $k_{rnw}$ are the relative permeabilities (Appendix B) of the brine water and CO_2_ in the fracture network, respectively. The viscosity at in situ conditions is $\mu_{w}$ for water and $\mu_{nw}$ for CO_2_. ϕ is the porosity of the fracture network. The source term is $f_{w}^{\prime }$ for water and $f_{nw}^{\prime }$ for CO_2_. $p_{bw}$ and $p_{nw}$ are the pore pressures of the brine water and CO_2_ in the fractures. $s_{bw}$ and $s_{nw}$ are the saturations of the brine water and CO_2_, respectively. H is the height in the vertical direction. $\rho_{nwa}$ and $\rho_{c}$ are the densities of the CO_2_ and the shale caprock under the standard conditions, respectively. g is the gravity acceleration.

The equation of the state gives the CO_2_ density, $\rho_{nw}$, as:(30)$$\rho_{nw}=\frac{M_{nw}}{Z_{nw}RT_{nw}}p_{nw}=\beta p_{nw}$$
where $M_{nw}$ and $Z_{nw}$ are the molecular weight and compressibility factor of CO_2_, respectively. R denotes the universal gas constant and $T_{nw}$ is the temperature of CO_2_. β is a constant of a range for the CO_2_ pressure and temperature. 

The pressure of CO_2_, $p_{nw}^{*}$, in the fractures is:(31)$$p_{nw}^{*}=s_{nw}p_{nw}$$

Introducing the relationship between the capillary pressure and saturation yields the final form of two-phase flow governing equations in the fractures.

For the brine water flow:(32)$$\phi C_{p}\frac{\partial p_{nw}}{\partial t}-\phi C_{p}\frac{\partial p_{bw}}{\partial t}+s_{bw}\frac{\partial \phi }{\partial t}=\nabla \cdot \left[\frac{kk_{rbw}}{\mu_{bw}}\left(\nabla p_{bw}+\rho_{bw}g\nabla H\right)\right]-\frac{dm_{bw}}{dt}+f_{bw}$$

For the CO_2_ flow:(33)$$\begin{array}{l} \phi^{\prime }\left(s_{nw}-p_{nw}C_{p}\right)\frac{\partial p_{nw}}{\partial t}+\phi^{\prime }p_{nw}C_{p}\frac{\partial p_{bw}}{\partial t}+\phi s_{nw}p_{nw}\frac{\partial \phi }{\partial t}\\ =\nabla \cdot \left[\frac{kk_{rnw}}{\mu_{nw}}p_{nw}\left(\nabla p_{nw}+\rho_{nw}g\nabla H\right)\right]-p_{a}\rho_{c}\frac{dm_{b}}{dt}+f_{nbw} \end{array}$$

The sorption modified porosity is: (34)$$\phi^{\prime }=\phi +\rho_{nwa}\rho_{c}\frac{V_{L}p_{L}}{{\left(p_{L}+p_{nw}^{*}\right)}^{2}}$$
(35)$$\phi =\frac{\Delta b}{b_{i}}=\left[1+n\frac{1-R_{m}}{\phi_{f0}}\right]\Delta \varepsilon_{e}$$
where $\frac{\partial \phi }{\partial t}$ is the change in the porosity with time under the action of a compaction, swelling/dehydration, sorption, and chemical reaction. $\left({-p}_{a}\rho_{c}\frac{dm_{b}}{dt}\right)$ is the source term of CO_2_. It is provided by the diffusion processes of free gas and absorption/adsorption process of adsorbed gas in the shale matrix (Appendix C). Similarly, $\left(-\rho_{bw}\frac{dm_{bw}}{dt}\right)$ is the water term. It is supplied by the dehydration of the matrix. $p_{a}$ corresponds to the atmospheric pressure. ${f^{\prime }}_{bw}=\rho_{bw}f_{bw}$ and ${f^{\prime }}_{nw}=\beta f_{nw}$.

### 4.2. Navier Equation for Shale Deformation

For the elastic shale saturated by the CO_2_ and brine water, the Navier equation for the deformation of the porous medium is [16]:(36)$$Gu_{i,jj}+\frac{G}{1-2\nu }u_{k,kj}=K\left(\varepsilon_{D,i}+\varepsilon_{s,i}\right)-\alpha {\bar{p}}_{,i}-f_{i}$$
where G is the shear modulus. $u_{i,jj}$ is the second-order derivative in the jth direction of the displacement in the *i*th direction $u_{i}$. ν is the Poisson’s ratio. ${\bar{p}}_{,i}$ is the derivative of the pore pressure in the ith direction and calculated by $\bar{p}=S_{w}p_{w}+S_{nw}P_{nw}$. α is the Biot coefficient. 

Equation (36) shows that there are four sources of the body force: the body force induced by dehydration-swelling ($K\varepsilon_{D,i}$), the body force induced by sorption-swelling ($K\varepsilon_{s,i}$), the friction force induced by the two-phase flow (drag force or $\alpha {\bar{p}}_{,i}$), and the body force induced by gravity ($f_{i}$). The resistance is decided by the Biot’s coefficient, and the body force induced by the swelling of the skeleton is correlated with the bulk modulus.

## 5. Numerical Modelling for the Assessment of Caprock Sealing Efficiency

### 5.1. Verification of This fully Coupled Multi-Physical Model

The British Geological Survey has carried out a gas breakthrough test on the argillite in the Callovo-Oxfordian formation [46]. Figure 6 shows the sample for this breakthrough test whose dimension was 5.39 cm high and 2.72 cm wide. For the flow field, there was no flow at the two side walls. For the deformation field, the two walls were also fixed. Helium was slowly injected from the bottom of the caprock with the pressure increasing from 6.5 to 10.5 MPa in a series of increments. A constant backpressure of 4.5 MPa was applied at the top of the caprock, and a confining stress of 12.5 MPa was maintained through the experiment. Before the gas injection, the caprock was fully saturated by water. Water and helium were allowed to flow out from the top boundary at a constant pressure. Gerard et al. [46] completed the hydro-mechanical modeling with a preferential gas pathway. They proposed a continuous finite element matrix embedding a single fracture to describe this problem. This study uses their computational parameters except the capillary-saturation relationship. We also use our log–log relationship instead of their Van Genuchten relationship for the relative permeabilities. Those parameters which were not given by Gerard et al. [46] are taken from other publications or our estimation. Table 1 and Table 2 summarize all the parameters in our computations. Because helium was used, no sorption and hydration were considered. In this proposed multi-physical model, the helium flow was only along the fracture network, thus no additional single fracture was required for the simulations. The flow rate of helium gas from the top boundary was calculated and compared with the experimental measurements. Our simulations considered two cases: the first case was a constant entry pressure of 2.1 MPa and the second case was a variable gas entry pressure which was evolving with the effective strain. The initial gas entry pressure was taken as $p_{ei}=2.1\;\mathrm{MPa}$.

Comparison between the experimental data and numerical simulations is shown in Figure 7. It is found that there are the same flow rates before the gas breakthrough in either the constant or variable entry pressure cases. These flow rates before the gas breakthrough are very low and depend on the absolute permeability and the initial gas saturation. With the gas–water front moving to the top boundary, the sample deforms and the fracture aperture thus changes, particularly near the mixing zone of helium and water. This deformation changes the gas entry pressure. The case of variable gas entry pressure is observed to have an earlier gas breakthrough time and the rapid increase in the flow rate after the gas breakthrough. The case of a constant gas entry pressure has a much later gas breakthrough time and a much lower flow rate after the gas breakthrough. It is also shown that a variable gas entry pressure can better reproduce the experimental data. A good agreement is observed between the numerical results by the variable gas entry pressure in Equation (20) and the experimental observations. There is a single fracture in the computational domain of their model. However, only the flow in fracture network is described in our model, thus a single fracture is not required. This treatment largely extends the capacity of our model to handle large-scale problems for the evaluation of the caprock sealing efficiency.

### 5.2. Impacts of Shale Matrix Dehydration on CO_2_ Penetration

#### 5.2.1. Model and Parameters

As shown in Figure 8, a typical geometric model of 10 m × 10 m was established for a one-dimensional penetration problem. For the flow field, there was no flow at the two side walls. However, CO_2_ and brine water can flow out from the top boundary. For the deformation field, the two walls and bottom were constrained. The CO_2_ was injected from the bottom with a given injection pressure. It increased from the reservoir pressure to 27 MPa in an exponential form and then remains constant for 1000 years. The diffusion time for both the CO_2_ and brine water was taken as 1d. The computational parameters are shown in Table 3. The caprock contains 54.1% quartz, 25.6% kaolinite, 13.5% illite and mica, and 2.5% K-feldspar by weight. The dehydration-induced swelling is still described by Equation (7). In this example, the short-term interaction mechanism was studied in the CO_2_–brine two-phase flow process. Furthermore, the impacts of the two-phase flow, the CO_2_ sorption, and dehydration/swelling on the caprock sealing efficiency were comprehensively analyzed in the two-phase flow process.

#### 5.2.2. Impacts of Matrix Dehydration on CO_2_–Brine Displacement Process

The impacts of dehydration-induced shrinkage on the CO_2_-water displacement are studied here. The two initial states of the shale matrix were assumed: a fully saturated with the dehydration (called dehydration) state and a fully saturated without the dehydration (called the base case) state. The time of the numerical calculation was 317 years (10^10^ s). Figure 9 presents the effect of the matrix dehydration on the water saturation in the fracture network when the injection time is 317 years. It is noted that the shrinkage strain due to the dehydration is only specified as 15 % of the swelling strain. This figure shows that the dehydration of the shale matrix has a slight impact on the water saturation distribution in the vertical direction. In the CO_2_–brine water two-phase flow process, two competitive factors result in the swelling or shrinkage of the shale matrix. Figure 10a presents the increase in the swelling strain induced by the CO_2_ sorption with time. The observation point is located at 0.1 m away from the lower face. The CO_2_ diffusion causes the swelling of the shale matrix. At the same time, the dehydration process makes the shale matrix shrink, as shown in Figure 10b. It is noted that the magnitude of the shrinkage strain is much smaller than the sorption-induced swelling strain in this example. The combination of the CO_2_ sorption-induced swelling and the dehydration-induced shrinkage makes the permeability ratio slightly increase, then decrease, and finally reach a low value. Figure 11 presents the permeability evolution at the observation point. The dehydration process contributes a little to the increase in the early stage but significantly reduces the extent of self-limiting in the fracture network. This reduction in self-limiting is also observed from the vertical distribution of the permeability ratio at time of 3.17 years in Figure 12. The minimum permeability ratio is bigger due to dehydration. This implies that the dehydration increases the fracture apertures and thus alleviates the reduction in the permeability of the fracture network due to swelling. The impact of the dehydration on the penetration depth is shown in Figure 13. In this sense, the penetration depth with the dehydration is much higher than without the dehydration. Therefore, the risk of a gas breakthrough is increased due to the dehydration.

#### 5.2.3. Self-Limiting Mechanism Analysis

The self-limiting mechanism of caprock is complicated. This mechanism comes from the two competitions of the shale matrix swellings: the CO_2_ sorption-induced swelling and the dehydration-induced shrinkage. As the flow paths narrow down and even close under the action of the swelling, the permeability in the fractures will decrease and the penetration speed of the CO_2_–water front will slow down [47]. If the action of the swelling becomes stronger along the flow direction, the efficiency of the caprock sealing can be enhanced. For the initially water-saturated shale matrix, the CO_2_ diffusion can induce the shale matrix dehydration [48]. This dehydration changes the water content and induces the shrinkage of the shale matrix. The fracture apertures get wider due to the shrinkage of the shale matrix during the CO_2_ diffusion. The worst case is to reopen the flow channels which are closing due to a full swelling of the shale matrix under in situ conditions. Such an increase in the apertures increases the permeability and penetration depth [35]. This condition reduces the efficiency of the caprock sealing and there is a potential leakage risk of CO_2_. Figure 12 shows that the permeability fluctuates and is different in the two-phase flow region or the sweeping zone which moves with the CO_2_–water front. After the two-phase flow, the permeability decreases by about 10% along the flow direction for the base case and approximately 6% from the contribution of dehydration. In this regard, the dehydration process has an effect on the swelling strain. In addition, the self-limiting/enhancing capacity has also been influenced. On this sense, it can be concluded that the strong dehydration within the caprock can significantly influence the caprock sealing. For the water-saturated caprock, the CO_2_ infiltration may cause the matrix dehydration [49]. Therefore, the caprock dehydration should be carefully considered in the evaluation of the caprock sealing.

## 6. Conclusions

A multi-physical coupling model was proposed to investigate the effects of dehydration and sorption on the efficiency of the caprock sealing. This model was further validated by a gas breakthrough test. The combined effects of CO_2_ adsorption-induced swelling and dehydration-induced shrinkage on the permeability and entry capillary pressure of the fracture network were studied through this model. Particularly, the impact of the dehydration-induced shrinkage on the penetration depth was particularly studied. The following conclusions can be drawn from these investigations:

First, this multi-physical coupling model is a sound tool for the assessment of the sealing efficiency of caprock. It includes the capacity of our previous model in illustrating the physical and mechanical properties of caprock, such as the compaction deformation, gas flow, and sorption. It also expands its capacity, including the dehydration of the shale matrix and the porosity and permeability evolution in the fracture network due to the dehydration shrinkage and compaction.

Second, the evolution mechanisms of the porosity and permeability are complicated and complex, particularly in the CO_2_–brine two-phase flow region and the gas sweeping region. These evolutions are the interaction results among the CO_2_ diffusion, mechanical compaction, two-phase flow, CO_2_ sorption-induced swelling, and dehydration-induced shrinkage. These interactions cause the effects of self-enhancing/limiting in these regions due to a swelling/shrinkage of the shale matrix. 

Finally, the sorption-induced swelling and dehydration-induced shrinkage in the saturated shale caprock are two competitive factors to alter the efficiency of caprock sealing. A CO_2_ infiltration may cause the matrix’s dehydration from the water-saturated caprock. This matrix dehydration can induce the re-opening of some fractures, enhance the permeability, and reduce the efficiency of caprock sealing, thus being a potential risk for CO_2_ geological sequestration. Caprock dehydration is worthy of being carefully considered in the evaluation of the caprock sealing efficiency.

## Figures and Tables

**Figure 1 ijerph-19-14574-f001:**
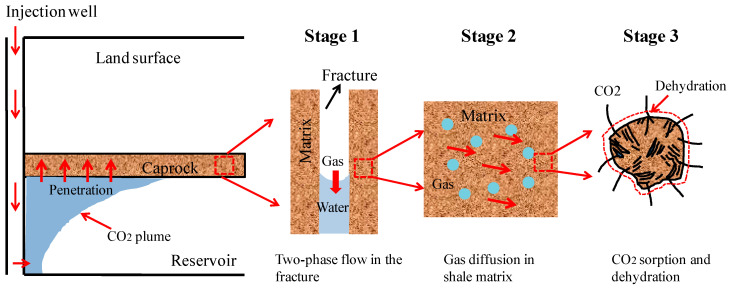
CO_2_ transport and dehydration mechanism of shale caprock.

**Figure 2 ijerph-19-14574-f002:**
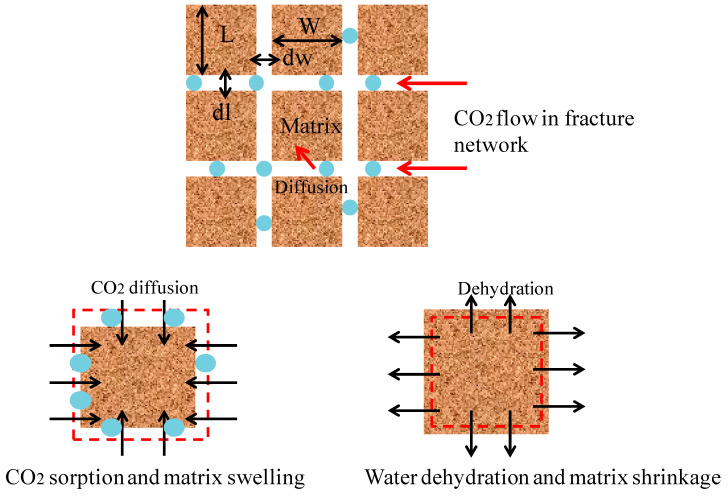
CO_2_ sorption and dehydration model in shale caprock. The abbreviated letters L, dl, W, and dw are short for length, derivative of length, width, and derivative of width, respectively.

**Figure 3 ijerph-19-14574-f003:**
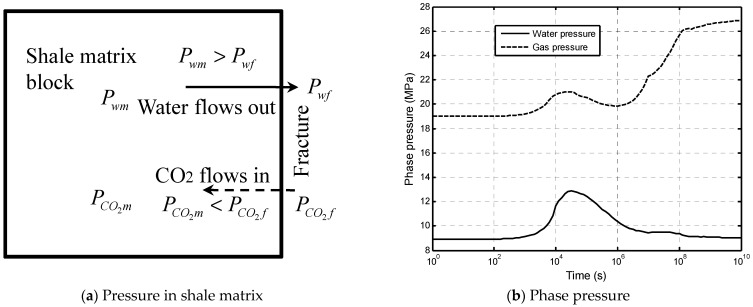
CO_2_-water displacement in tight caprock. The abbreviated letters $P_{wm}$, $P_{wf}$, $P_{CO_{2}m}$, $P_{CO_{2}f}$ are short for the pressures of water in the matrix, water in the fractures, CO_2_ in the matrix, and CO_2_ in the fractures, respectively.

**Figure 4 ijerph-19-14574-f004:**
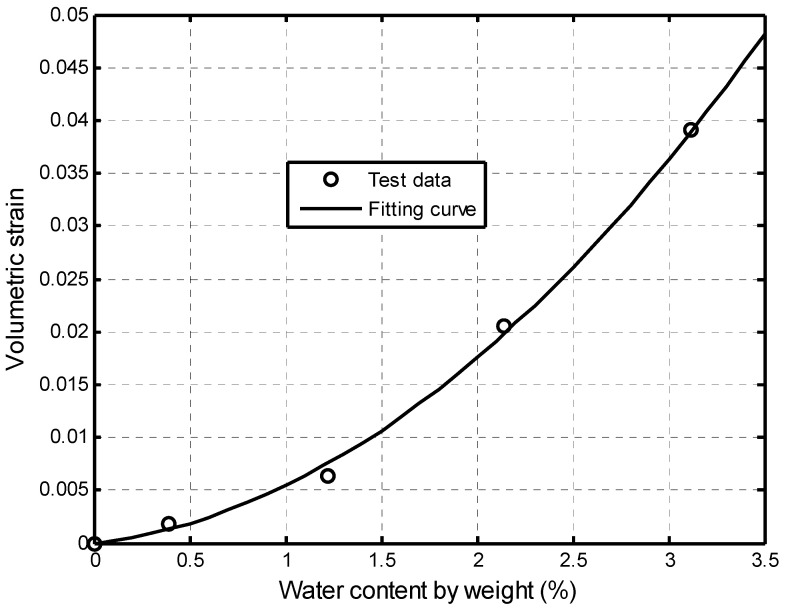
Hydration-induced strain in Mancos shale [42].

**Figure 5 ijerph-19-14574-f005:**
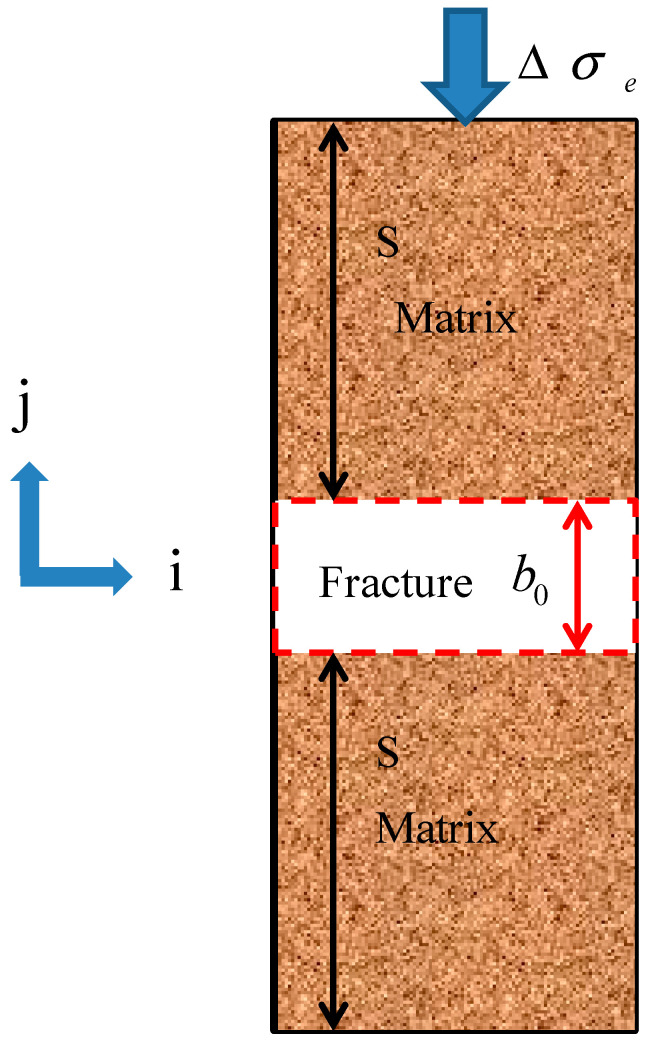
Deformation of fractured shale element.

**Figure 6 ijerph-19-14574-f006:**
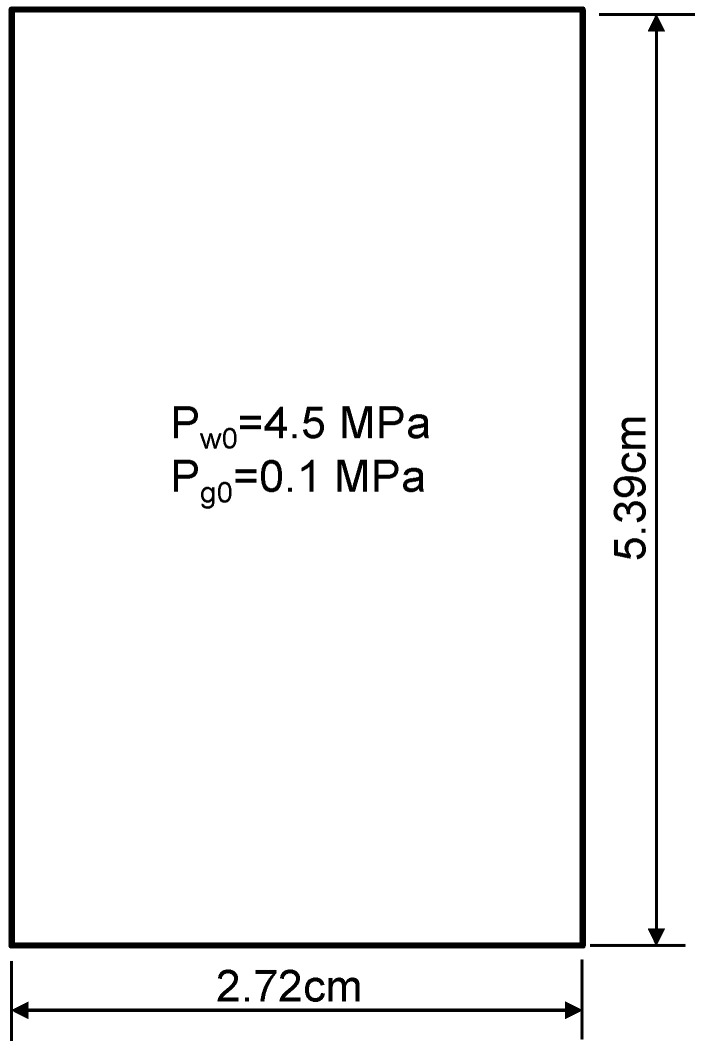
Geometry of breakthrough test problem.

**Figure 7 ijerph-19-14574-f007:**
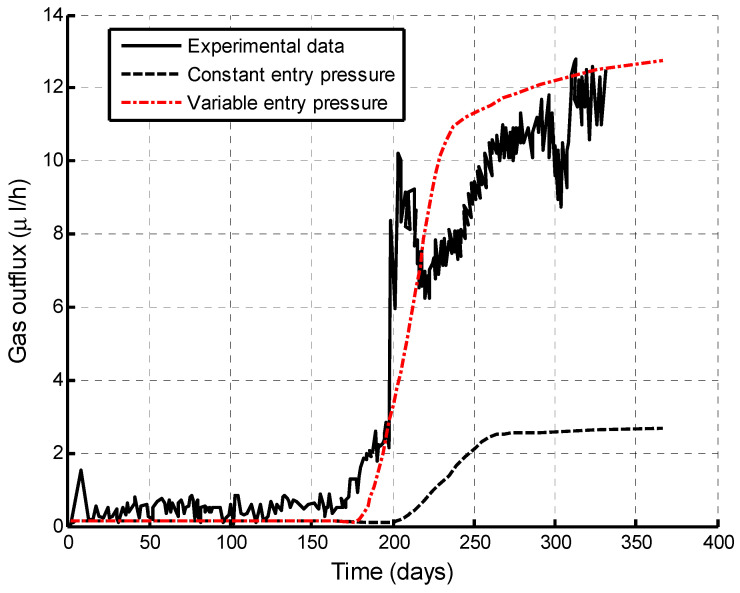
Comparison of gas outflux between experimental data [46] and our numerical simulations.

**Figure 8 ijerph-19-14574-f008:**
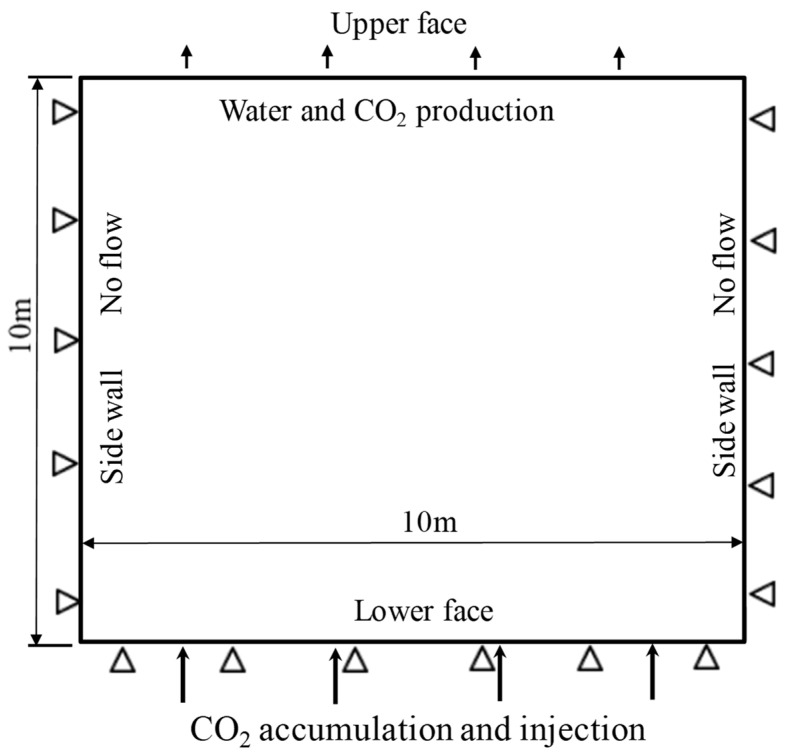
Numerical model for CO_2_-water two-phase flow in caprock.

**Figure 9 ijerph-19-14574-f009:**
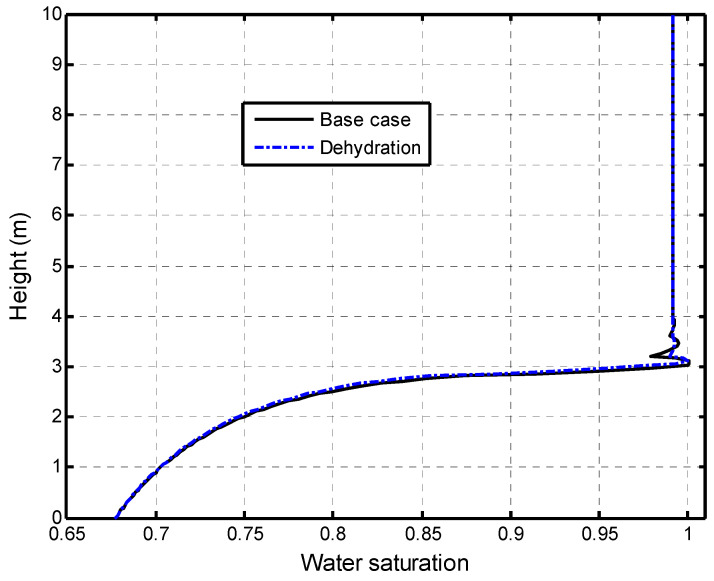
Effect of dehydration on the water saturation in fracture network at 317 years.

**Figure 10 ijerph-19-14574-f010:**
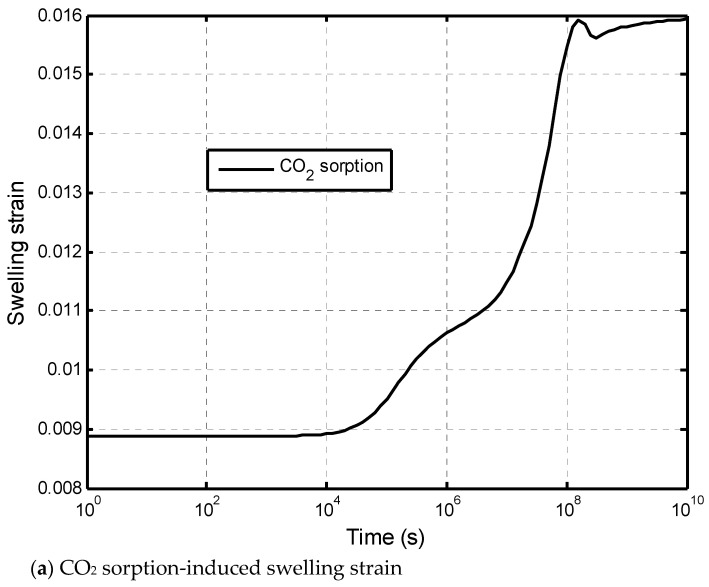
Swelling/shrinkage strain induced by two competitive factors at the observation point.

**Figure 11 ijerph-19-14574-f011:**
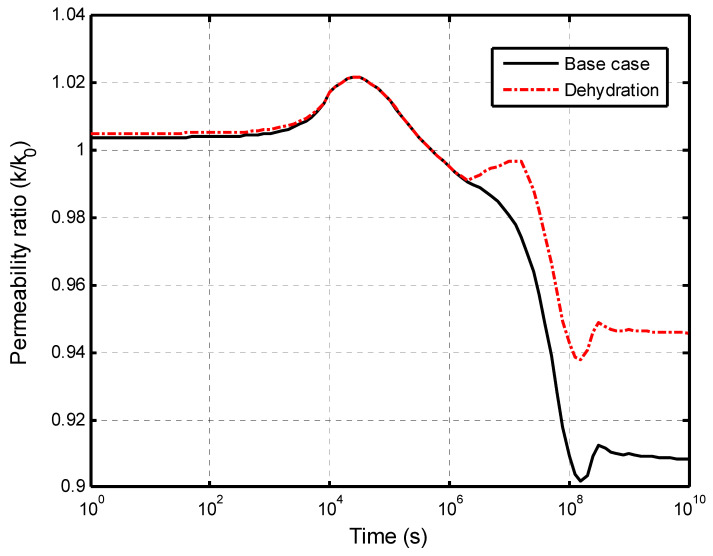
Comparations of permeability ratio at the observation point.

**Figure 12 ijerph-19-14574-f012:**
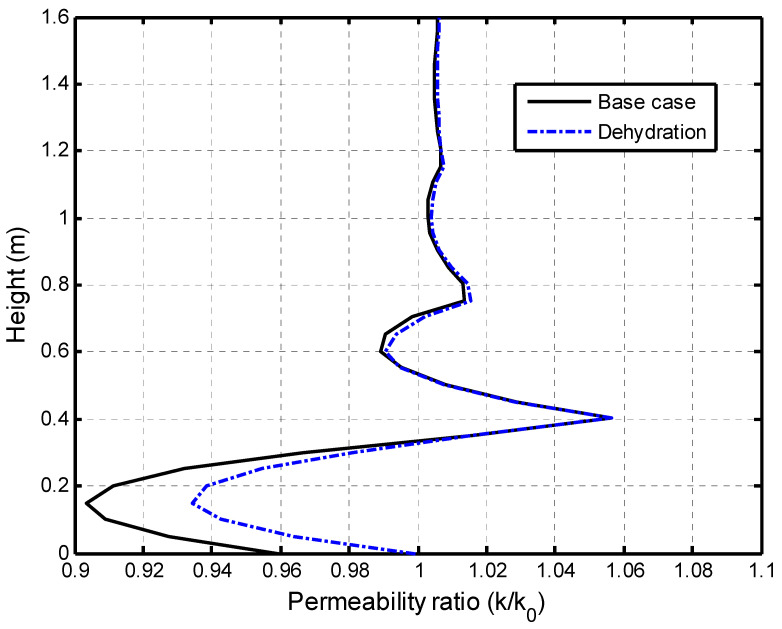
Comparations of permeability distribution in the vertical direction at 3.17 years.

**Figure 13 ijerph-19-14574-f013:**
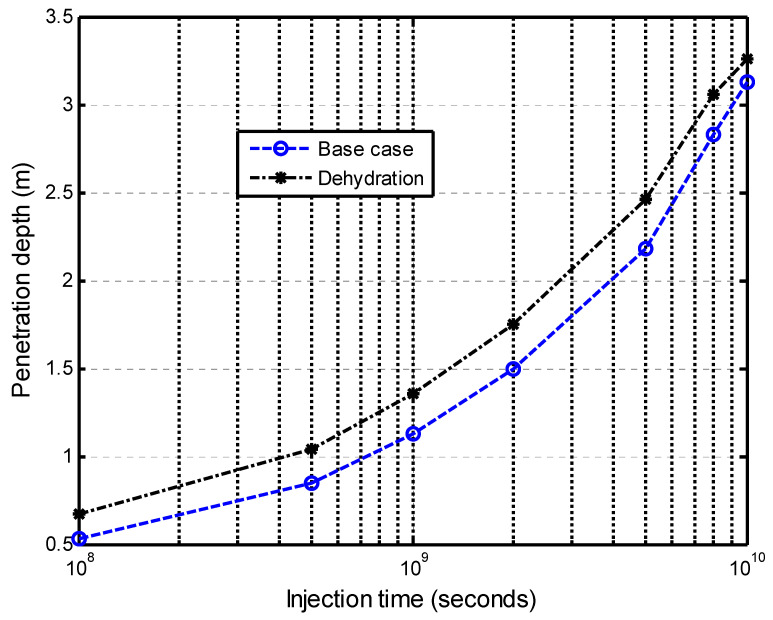
Effect of dehydration on the penetration depth.

**Table 1 ijerph-19-14574-t001:** Langmuir pressure and strain in gas breakthrough test.

Relative Constant	Directions to the Bedding		
	Paralleling	Perpendicular	Average
$\mathrm{Langmuir}\;\mathrm{pressure}\;p_{L}$ (MPa)	6.5	6.2	6.0
$\mathrm{Langmuir}\;\mathrm{strain}\;\varepsilon_{L}\;\left(\%\right)$	1.5	3.7	2.7

**Table 2 ijerph-19-14574-t002:** Computation parameters for gas breakthrough test.

Parameter	Unit	Value	Physical Meanings
$s_{nwr}$		0.05	Helium residual saturation
$s_{bwr}$		0.6	Water residual saturation
$P_{ei}$	MPa	2.1	Initial entry capillary pressure
$\mu_{bw}$	Pa*s	0.00085	Water viscosity
$\mu_{nw}$	Pa*s	$2.0\times 10^{-5}$	Helium viscosity
$\lambda_{bw}$		3	Water’s Corey parameter
$\lambda_{nw}$		3	Helium’s Corey parameter
λ		1.1	Pore size distribution index
T	K	300	Experimental temperature
$p_{bw0}$	MPa	4.5	Initial pressure of water
$p_{nw0}$	MPa	6.66	Initial pressure of helium
$\phi_{0}$		0.18	Initial porosity
$k_{0}$	m^2^	$1.33\times 10^{-20}$	Initial shale permeability
$p_{L}$	MPa	6	Langmuir pressure of helium
$E_{c}$	GPa	3.8	Overall Young’s modulus of shale
$E_{s}$	GPa	9.5	Matrix Young’s modulus of shale
ν		0.3	Poisson’s ratio
$\rho_{c}$	kg/m^3^	2300	Density of shale
$k_{nwe}$		0.005	Helium’s relative permeability at end point
$p_{nwout}$	MPa	6.62	Outlet pressure of helium
$p_{bwout}$	MPa	4.5	Outlet pressure of water

**Table 3 ijerph-19-14574-t003:** Computation parameters for CO_2_ penetration.

Parameter	Unit	Value	Physical Meanings
$s_{nwr}$		0.15	CO_2_ residual saturation
$s_{bwr}$		0.6	Brine water residual saturation
$P_{ei}$	MPa	10	Capillary entry pressure at initial state
T	K	353.15	Temperature taken from CO_2_ storage reservoirs
$\mu_{bw}$	Pa*s	$3.6\times 10^{-4}$	Viscosity of brine water
$\mu_{nw}$	Pa*s	$5.2\times 10^{-5}$	Viscosity of CO_2_
$\lambda_{bw}$		6.5	Brine Water’s Corey parameter
$\lambda_{nw}$		2.6	CO_2_’s Corey parameter
λ		2.0	Distribution index of pore size
$p_{bwf}$	MPa	8.95	Pressure at the top boundary
$p_{bw0}$	MPa	8.95	Water pressure at initial state
$p_{nw0}$	MPa	19	CO_2_ pressure at initial state
$\phi_{0}$		0.04	Initial porosity
$k_{0}$	m^2^	$1.5\times 10^{-19}$	Initial permeability
$p_{L}$	MPa	6	CO_2_ Langmuir pressure
$V_{L}$	m^3^/kg	0.03	Shale Langmuir sorption capacity
$E_{c}$	GPa	8	Overall Young’s modulus of shale
$E_{s}$	GPa	20	Matrix Young’s modulus of shale
ν		0.30	Poisson’s ratio
$\rho_{c}$	kg/m^3^	2300	Density of shale
$p_{nwout}$	MPa	19	Outlet pressure of CO_2_
$p_{bwout}$	MPa	8.95	Outlet pressure of brine water
D	m^2^/s	$1.2\times 10^{-11}$	Coefficient of diffusion in shale
$k_{rbw}^{\max}$		1.0	Brine water relative permeability at end point
$k_{rnw}^{\max}$		0.015	CO_2_ relative permeability at end point

## Data Availability

Not applicable.

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
