# Peer review of "Combined Effects of CO2 Adsorption-Induced Swelling and Dehydration-Induced Shrinkage on Caprock Sealing Efficiency"

_ijerph, 2022, doi:10.3390/ijerph192114574_

Round 1

Reviewer 1 Report

Combined effects of CO2 adsorption-induced swelling and dehydration-induced shrinkage on caprock sealing efficiency are investigated in this manuscript. The authors mention transport mechanisms can be adapted from the combination of two-phase flow in the fractures, CO2 sorption and diffusion in the matrix, and matrix dehydration. Also, it can be exerted the caprock sealing's stability is modified by CO2 sorption and dehydration-induced swelling or shrinkage of the shale matrix. It means that CO2 sorption has an important impact on the stability of shale- or clay-caprock sealing. This try can be evaluated concerning CO2 adsorption and Caprock stability. However, this version presents a full-messy article that may confuse the readers with what the authors follow. Methods and results are shifted improperly, and data are shown from experimental tests without introducing them in Methods! From Conclusion, the authors state, "It includes the capacity of our previous model in illustrating the physical and mechanical properties of caprock such as compaction deformation." Still, it is unclear which the previous model. Overall, this version must be shortened and written correctly. 

Some comments:

A polish in syntax is needed to present a fluent-readable article. E.g., L. 69, Change from to by.

Provide a valid reference for the following statement: "If the CO2 pressure is lower than the sum of the reservoir pressure and entry capillary pressure, the Darcy flow does not occur."

Most parts of 2.1 are redundant, generally stating multi-phase interaction. 

Is 2.1 a model adopted or introduced? Unclear! 

Indicate what abbreviated letters in Fig. 2 are.

This is an irrelative statement: "Shale is a sedimentary rock formed by dewatering and cementation of clay minerals, and the clay minerals are water-sensitive."

While describing the model, state the statements in the past tense. E.g., were instead of are in: "Both the fractures and matrix were assumed to have the same initial phase pressures of CO2-brine water two-phase flow."

Fig. 3 needs to be clarified further in terms of abbreviations, presenting a legend, or in its caption.

Eq. 5: the authors put constants without addressing more about them. Refer to A, Alpha, and Alpha1.

From eq. 7 and fig. 4, what is the unit for volumetric strain?

S and S0 are undefined in Eqs. 8 and 9. 

It seems Eqs. introduced here are adopted from publications, and citations should be considered accordingly.

The authors mention the Navier equation in eq. 36 without referencing it. 

The authors have improperly shifted two main sections of Models and Results. 5.1 shows this messy as well.

L. 382-383: I could not find how the authors compare the modeling results with the experimental tests without introducing the laboratory method. 

Also, the following statement can be replaced instead of what the current version shows:

"CO2 sorption has an important impact on the stability of caprock sealing. The results are variable with organic compositions, types, and shale contents. Furthermore, clay minerals in shale layers can adsorb some free gas. The gas capacity of adsorption varies with clay type, volume, and type of gas itself. The adsorbed volume of gas is closely related to the pore's surface area. It is found that there are similar sorption behaviors between coal and shale."

Author Response

Please see the attchment

Reviewer 2 Report

Minor drawbacks and recommended improvements

1

Line 29, page 1

The Keywords dehydration-induced shrinkage; caprock sealing efficiency is in the title. I suggest that the authors remove these keywords from the title or keywords and replace them with others.

2

page 130, 137, 143, 151, 152, 467, 471, 481, 482-487.

The authors should indicate updated references to other studies with similar results to corroborate these results.

3

page 432, 481, 432

The authors should replace the word “obviously” with “in this sense” or “In this regard.”

Author Response

Please see the word file attached.

Round 2

Reviewer 1 Report

The authors have addressed to comments from the previous version accordingly.